# RETHINKING NETWORK DESIGN AND LOCAL GEOMETRY IN POINT CLOUD: A SIMPLE RESIDUAL MLP FRAMEWORK

**Xu Ma[1], Can Qin[1], Haoxuan You[2], Haoxi Ran[1], Yun Fu[1]**
[1]Northeastern University, Boston, MA, USA
[2]Columbia University, New York, NY, USA
`{ma.xu1,qin.ca,ran.h}@northeastern.edu`
`{haoxuanyou,ranhaoxi}@gmail.com`
`yunfu@ece.neu.edu`

## ABSTRACT

Point cloud analysis is challenging due to irregularity and unordered data structure. To capture the 3D geometries, prior works mainly rely on exploring sophisticated local geometric extractors using convolution, graph, or attention mechanisms. These methods, however, incur unfavorable latency during inference, and the performance saturates over the past few years. In this paper, we present a novel perspective on this task. We notice that detailed local geometrical information *probably is not* the key to point cloud analysis – we introduce a *pure residual MLP* network, called PointMLP, which integrates no "sophisticated" local geometrical extractors but still performs very competitively. Equipped with a proposed lightweight geometric affine module, PointMLP delivers the new state-of-the-art on multiple datasets. On the real-world ScanObjectNN dataset, our method even surpasses the prior best method by **3.3**% accuracy. We emphasize that PointMLP achieves this strong performance *without* any sophisticated operations, hence leading to a superior inference speed. Compared to most recent CurveNet, PointMLP **trains 2× faster, tests 7× faster**, and is more accurate on ModelNet40 benchmark. We hope our PointMLP may help the community towards a better understanding of point cloud analysis. The code is available at https://github.com/ma-xu/pointMLP-pytorch.

## 1 INTRODUCTION

Lately, point cloud analysis has emerged as a popular topic in 3D understanding, attracting attention from academia and industry (Qi et al., 2017a; Shi et al., 2019; Xu et al., 2020). Different from 2D images represented by regular dense pixels, point clouds are composed of unordered and irregular sets of points $\mathcal{P} \in \mathbb{R}^{N \times 3}$, making it infeasible to apply image processing methods to point cloud analysis directly. Meanwhile, the nature of sparseness and the presence of noises further restrict the performance. In the past few years, endowing with neural networks, point cloud analysis has seen a great improvement in various applications, including 3D shape classification (Qi et al., 2017a), semantic segmentation (Hu et al., 2020) and object detection (Shi & Rajkumar, 2020), etc.

Recent efforts have shown promising results for point cloud analysis by exploring local geometric information, using convolution (Li et al., 2021a), graph (Li et al., 2021a), or attention mechanism (Guo et al., 2021) (see Section 2 for details). These methods, despite their gratifying results, have mainly relied on the premise that an elaborate local extractor is essential for point cloud analysis, leading to the competition for careful designs that explore fine local geometric properties. Nevertheless, sophisticated extractors are not without drawbacks. On the one hand, due to prohibitive computations and the overhead of memory access, these sophisticated extractors hamper the efficiency of applications in natural scenes. As an example, until now, most 3D point cloud applications are still based on the simple PointNet ( and PointNet++) or the voxel-based methods (Liu et al., 2021; Li et al., 2021b; Zhang et al., 2021). However, applications that employ the aforementioned advanced methods are rare in literature. On the other hand, the booming sophisticated extractors saturate the

performance since they already describe the local geometric properties well. A more complicated design is no longer to improve the performance further. These phenomena suggest that we may need to stop the race of local feature extraction designing, rethinking the necessity of elaborate local feature extractors and further revisiting the succinct design philosophy in point cloud analysis.

In this paper, we aim at the ambitious goal of building a deep network for point cloud analysis using only residual feed-forward MLPs, without any delicate local feature explorations. By doing so, we eschew the prohibitive computations and continued memory access caused by the sophisticated local geometric extractors and enjoy the advantage of efficiency from the highly-optimized MLPs. To further improve the performance and generalization ability, We introduce a lightweight local geometric affine module that adaptively transforms the point feature in a local region. We term our new network architecture as PointMLP. In the sense of MLP-based design philosophy, our PointMLP is similar to PointNet and PointNet++ (Qi et al., 2017a;b). However, our model is more generic and exhibits promising performance. Different from the models with sophisticated local geometric extractors (e.g., DeepGCNs (Li et al., 2019), RSCNN (Liu et al.,

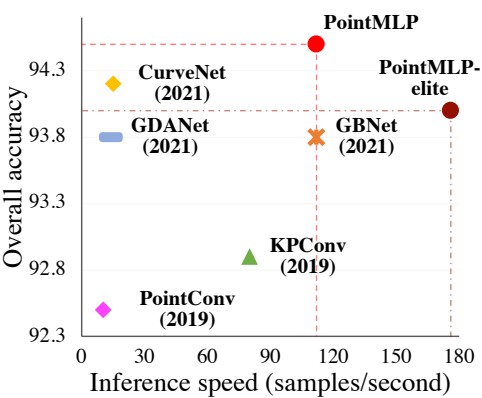

Figure 1: Accuracy-speed tradeoff on Model-Net40. Our PointMLP performs best. Please refer to Section 4 for details.

2019b), RPNet (Ran et al., 2021).), our PointMLP is conceptually simpler and achieves results on par or even better than these state-of-the-art methods (see Figure 1). Keep in mind that we did not challenge the advantages of these local geometric extractors and we acknowledge their contributions; however, a more succinct framework should be studied considering both the efficiency and accuracy. In Table 1, we systemically compare our PointMLP with some representative methods.

Even though the design philosophy is simple, PointMLP (as well as the elite version) exhibits superior performance on 3D point cloud analysis. Specifically, we achieve the state-of-the-art classification performance, **94.5%**, on the ModelNet40 benchmark, and we outperform related works by **3.3%** accuracy on the real-world ScanObjectNN dataset, with a significantly higher inference speed.

## 2 RELATED WORK

**Point cloud analysis.** There are mainly two streams to process point cloud. Since the point cloud data structure is irregular and unordered, some works consider projecting the original point clouds to intermediate voxels (Maturana & Scherer, 2015; Shi et al., 2020) or images (You et al., 2018; Li et al., 2020), translating the challenging 3D task into a well-explored 2D image problem. In this regime, point clouds understanding is largely boosted and enjoys the fast processing speed from 2D images or voxels. Albeit efficient, information loss caused by projection degrades the representational quality of details for point clouds (Yang et al., 2019). To this end, some methods are proposed to process the original point cloud sets directly. PointNet (Qi et al., 2017a) is a pioneering work that directly consumes unordered point sets as inputs using shared MLPs. Based on PointNet, PointNet++ (Qi et al., 2017b) further introduced a hierarchical feature learning paradigm to capture the local geometric structures recursively. Owing to the local point representation (and multi-scale information), PointNet++ exhibits promising results and has been the cornerstone of modern point cloud methods (Wang et al., 2019; Fan et al., 2021; Xu et al., 2021a). Our PointMLP also follows the design philosophy of PointNet++ but explores a simpler yet much deeper network architecture.

**Local geometry exploration.** As PointNet++ built the generic point cloud analysis network framework, the recent research focus is shifted to how to generate better regional points representation. Predominantly, the explorations of local points representation can be divided into three categories: convolution-, graph-, and attention-based methods. One of the most distinguished convolution-based methods is PointConv (Wu et al., 2019). By approximating continuous weight and density functions in convolutional filters using an MLP, PointConv is able to extend the dynamic filter to a new convolution operation. Also, PAConv (Xu et al., 2021a) constructs the convolution

kernel by dynamically assembling basic weight matrices stored in a weight bank. Without modifying network configurations, PAConv can be seamlessly integrated into classical MLP-based pipelines. Unlike convolution-based methods, Graph-based methods investigate mutually correlated relationships among points with a graph. In Wang et al. (2019), an EdgeConv is proposed to generate edge features that describe the relationships between a point and its neighbors. By doing so, a local graph is built, and the point relationships are well preserved. In 3D-GCN (Lin et al., 2021), authors aim at deriving deformable 3D kernels using a 3D Graph Convolution Network. Closely related to graph-based methods, the attention-based methods exhibit excellent ability on relationship exploration as well, like PCT (Guo et al., 2021) and Point Transformer (Zhao et al., 2021; Engel et al., 2020). With the development of local geometry exploration, the performances on various tasks appear to be saturated. Continuing on this track would bring minimal improvements. In this paper, we showcase that even without the carefully designed operations for local geometry exploration, a pure deep hierarchical MLP architecture is able to exhibit gratifying performances and even better results.

**Deep network architecture for point cloud.** Interestingly, the development of point cloud analysis is closely related to the evolution of the image processing network. In the early era, works in the image processing field simply stack several learning layers to probe the performance limitations (Krizhevsky et al., 2012; Simonyan & Zisserman, 2015; Dong et al., 2014). Then, the great success of deep learning was significantly promoted by deep neural architectures like ResNet (He et al., 2016), which brings a profound impact to various research fields. Recently, attention-based models, including atten-

Table 1: Systematic comparison among some representative methods. "Deep" indicates that a model is expandable along depth. "Opt." stands for the principal operator.

| Method | hierarchy | locality | deep | opt. |
|---|---|---|---|---|
| PointNet | ✗ | ✗ | ✗ | MLP |
| PointNet++ | ✓ | ✓ | ✗ | MLP |
| DGCNN | ✗ | ✓ | ✗ | GCN |
| DeepGCNs | ✓ | ✓ | ✓ | GCN |
| PointConv | ✓ | ✓ | ✗ | Conv. |
| Point Trans. | ✓ | ✓ | ✓ | Atten. |
| PointMLP | ✓ | ✓ | ✓ | MLP |

tion blocks (Wang et al., 2018) and Transformer architectures (Dosovitskiy et al., 2021), further flesh out the community. Most recently, the succinct deep MLP architectures have attracted a lot of attention due to their efficiency and generality. Point cloud analysis follows the same develop history as well, from MLP-based PointNet (Qi et al., 2017a), deep hierarchical PointNet++ (Qi et al., 2017b), convolution-/graph-/relation- based methods (Wu et al., 2019; Wang et al., 2019; Ran et al., 2021), to state-of-the-art Transformer-based models (Guo et al., 2021; Zhao et al., 2021). In this paper, we abandon sophisticated details and present a simple yet effective deep residual MLP network for point cloud analysis. Instead of following the tendency in the vision community deliberately, we are in pursuit of an inherently simple and empirically powerful architecture for point cloud analysis.

## 3 DEEP RESIDUAL MLP FOR POINT CLOUD

We propose to learn the point cloud representation by a simple feed-forward residual MLP network (named PointMLP), which hierarchically aggregates the local features extracted by MLPs, and abandons the use of delicate local geometric extractors. To further improve the robustness and improve the performance, we also introduce a lightweight geometric affine module to transform the local points to a normal distribution. The detailed framework of our method is illustrated in Figure 2.

### 3.1 REVISITING POINT-BASED METHODS

The design of point-based methods for point cloud analysis dates back to the PointNet and PointNet++ papers (Qi et al., 2017a;b), if not earlier. The motivation behind this direction is to directly consume point clouds from the beginning and avoid unnecessary rendering processes.

Given a set of points $\mathcal{P} = \{p_i | i = 1, \cdots, N\} \in \mathbb{R}^{N \times 3}$, where $N$ indicates the number of points in a $(x, y, z)$ Cartesian space, point-based methods aims to directly learn the underlying representation $f$ of $\mathcal{P}$ using neural networks. One of the most pioneering works is PointNet++, which learns hierarchical features by stacking multiple learning stages. In each stage, $N_s$ points are re-sampled by

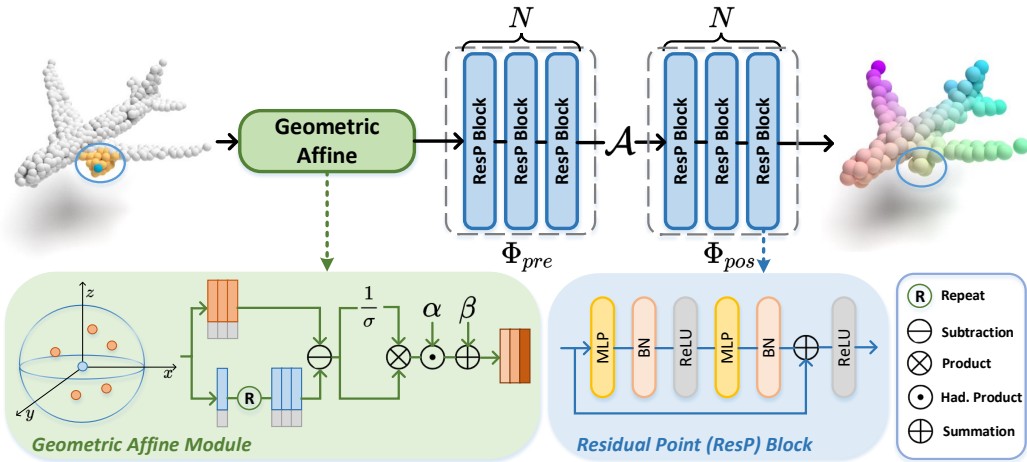

Figure 2: Overview of one stage in PointMLP. Given an input point cloud, PointMLP progressively extracts local features using residual point MLP blocks. In each stage, we first transform the local points using a geometric affine module, then they are extracted before and after the aggregation operation, respectively. PointMLP progressively enlarges the receptive field and models complete point cloud geometric information by repeating multiple stages.

the farthest point sampling (FPS) algorithm where $s$ indexes the stage and $K$ neighbors are employed for each sampled point and aggregated by max-pooling to capture local structures. Conceptually, the kernel operation of PointNet++ can be formulated as:

$$g_i = \mathcal{A}\left(\Phi\left(f_{i,j}\right)|j = 1, \cdots, K\right), \tag{1}$$

where $\mathcal{A}\left(\cdot\right)$ means aggregation function (max-pooling in PointNet++), $\Phi\left(\cdot\right)$ denotes the local feature extraction function (MLP in PointNet++), and $f_{i,j}$ is the $j$-th neighbor point feature of $i$-th sampled point. By doing so, PointNet++ is able to effectively capture local geometric information and progressively enlarge the receptive fields by repeating the operation.

In the sense of network architecture design, PointNet++ exhibits a universal pipeline for point cloud analysis. Following this pipeline, some plug-and-play methods have been proposed, mainly focusing on the local feature extractor $\Phi\left(\cdot\right)$ (Xu et al., 2021a; Liu et al., 2019b; Thomas et al., 2019; Zhao et al., 2021). Generally, these local feature extractors thoroughly explore the local geometric information using convolution, graph, or self-attention mechanisms. In RSCNN (Liu et al., 2019b), the extractor is mainly achieved by exploring point relations as follow:

$$\Phi\left(f_{i,j}\right) = \mathrm{MLP}\left(\left[\|x_{i,j} - x_i\|_2, x_{i,j} - x_i, x_{i,j}, x_i\right]\right) * f_{i,j}, \forall j \in \{1, \cdots, K\}, \tag{2}$$

where $[\cdot]$ is the concatenation operation and MLP is a small network composed of a Fully-connected (FC) layer, Batch Normalization layer, and activation function. Unlike RSCNN, Point Transformer introduces the self-attention mechanism into point cloud analysis and considers the similarities between pair-wise points in a local region. To this end, it re-formulates the extractor as:

$$\Phi\left(f_i\right) = \sum_{j=1}^{k} \rho\left(\gamma\left(\varphi\left(f_i\right) - \psi\left(f_{i,j}\right) + \delta\right)\right) \odot \left(\alpha\left(f_{i,j} + \delta\right)\right), \tag{3}$$

where $\gamma, \varphi, \psi$ and $\alpha$ are linear mapping function, "$\odot$" is a Hadamard product, and $\rho$ is a soft-max normalization. In particular, Point Transformer introduces a relative position encoding, $\delta = \theta\left(x_i - x_{i,j}\right)$, where the relative position is encoded by two FC layers with a ReLU non-linearity layer, into both attention weights and features. The lightweight positional encoder largely improves the performance of Point Transformer.

While these methods can easily take the advantage of detailed local geometric information and usually exhibit promising results, two issues limit their development. First, with the introduction of delicate extractors, the computational complexity is largely increased, leading to prohibitive infer-

ence latency [1]. For example, the FLOPs of Equation 3 in Point Transformer would be $14Kd^2$, ignoring the summation and subtraction operations. Compared with the conventional FC layer that enjoys $2Kd^2$ FLOPs, it increases the computations by times. Notice that the memory access cost is not considered yet. Second, with the development of local feature extractors, the performance gain has started to saturate on popular benchmarks. Moreover, empirical analysis in Liu et al. (2020) reveals that most sophisticated local extractors make surprisingly similar contributions to the network performance under the same network input. Both limitations encourage us to develop a new method that circumvents the employment of sophisticated local extractors, and provides gratifying results.

## 3.2 FRAMEWORK OF POINTMLP

In order to get rid of the restrictions mentioned above, we present a simple yet effective MLP-based network for point cloud analysis that no sophisticated or heavy operations are introduced. The key operation of our PointMLP can be formulated as:

$$g_i = \Phi_{pos}\left(\mathcal{A}\left(\Phi_{pre}\left(f_{i,j}\right), |j = 1, \cdots, K\right)\right), \tag{4}$$

where $\Phi_{pre}\left(\cdot\right)$ and $\Phi_{pos}\left(\cdot\right)$ are residual point MLP blocks: the shared $\Phi_{pre}\left(\cdot\right)$ is designed to learn shared weights from a local region while the $\Phi_{pos}\left(\cdot\right)$ is leveraged to extract deep aggregated features. In detail, the mapping function can be written as a series of homogeneous residual MLP blocks, $\text{MLP}\left(x\right) + x$, in which MLP is combined by FC, normalization and activation layers (repeated two times). Following Qi et al. (2017a), we consider the aggregation function $\mathcal{A}\left(\cdot\right)$ as max-pooling operation. Equation 4 describes one stage of of PointMLP. For a hierarchical and deep network, we recursively repeat the operation by $s$ stages. Albeit the framework of PointMLP is succinct, it exhibits some prominent merits. 1) Since PointMLP only leverages MLPs, it is naturally invariant to permutation, which perfectly fits the characteristic of point cloud. 2) By incorporating residual connections, PointMLP can be easily extended to dozens layers, resulting deep feature representations. 3) In addition, since there is no sophisticated extractors included and the main operation is only highly optimized feed-forward MLPs, even we introduce more layers, our PointMLP still performs efficiently. Unless explicitly stated, the networks in our experiments use four stages, and two residual blocks in both $\Phi_{pre}\left(\cdot\right)$ and $\Phi_{pos}\left(\cdot\right)$. We employ k-nearest neighbors algorithm (kNN) to select the neighbors and set the number $K$ to 24.

## 3.3 GEOMETRIC AFFINE MODULE

While it may be easy to simply increase the depth by considering more stages or stacking more blocks in $\Phi_{pre}$ and $\Phi_{pos}$, we notice that a simple deep MLP structure will decrease the accuracy and stability, making the model less robust. This is perhaps caused by the sparse and irregular geometric structures in local regions. Diverse geometric structures among different local regions may require different extractors but shared residual MLPs struggle at achieving this. We flesh out this intuition and develop a lightweight geometric affine module to tackle this problem. Let $\{f_{i,j}\}_{j=1,\cdots,k} \in \mathbb{R}^{k \times d}$ be the grouped local neighbors of $f_i \in \mathbb{R}^d$ containing $k$ points, and each neighbor point $f_{i,j}$ is a $d$-dimensional vector. We transform the local neighbor points by the following formulation:

$$\{f_{i,j}\} = \alpha \odot \frac{\{f_{i,j}\} - f_i}{\sigma + \epsilon} + \beta, \quad \sigma = \sqrt{\frac{1}{k \times n \times d} \sum_{i=1}^{n} \sum_{j=1}^{k} (f_{i,j} - f_i)^2}, \tag{5}$$

where $\alpha \in \mathbb{R}^d$ and $\beta \in \mathbb{R}^d$ are learnable parameters, $\odot$ indicates Hadamard production, and $\epsilon = 1e^{-5}$ is a small number for numerical stability (Ioffe & Szegedy, 2015; Wu & He, 2018; Dixon & Massey Jr, 1951). Note that $\sigma$ is a scalar describes the feature deviation across all local groups and channels. By doing so, we transform the local points to a normal distribution while maintaining original geometric properties.

## 3.4 COMPUTATIONAL COMPLEXITY AND ELITE VERSION

Although the FC layer is highly optimized by mainstream deep learning framework, the theoretical number of parameters and computational complexity are still high. To further improve the efficiency,

---

[1]We emphasize that the model complexity could not be simply revealed by FLOPs or parameters, other metrics like memory access cost (MAC) and the degree of parallelism also significantly affect the speed (Ma et al., 2018; Zhang et al., 2020). However, these important metrics are always ignored in point clouds analysis.

Table 2: Classification results on ModelNet40 dataset. With only 1k points, our method achieves state-of-the-art results on both class mean accuracy (mAcc) and overall accuracy (OA) metrics. We also report the speed of some open-sourced methods by samples/second tested on one Tesla V100-pcie GPU and four cores AMD EPYC 7351@2.60GHz CPU. * For KPConv, we take the results from the original paper. The best is marked in **bold** and second best is in blue.

| Method | Inputs | mAcc(%) | OA(%) | Param. | Train speed | Test speed |
|---|---|---|---|---|---|---|
| PointNet (Qi et al., 2017a) | 1k P | 86.0 | 89.2 | | | |
| PointNet++ (Qi et al., 2017b) | 1k P | - | 90.7 | 1.41M | **223.8** | **308.5** |
| PointNet++ (Qi et al., 2017b) | 5k P+N | - | 91.9 | 1.41M | | |
| PointCNN (Li et al., 2018b) | 1k P | 88.1 | 92.5 | | | |
| PointConv (Wu et al., 2019) | 1k P+N | - | 92.5 | 18.6M | 17.9 | 10.2 |
| KPConv (Thomas et al., 2019) | 7k P | - | 92.9 | 15.2M | 31.0* | 80.0* |
| DGCNN (Wang et al., 2019) | 1k P | 90.2 | 92.9 | | | |
| RS-CNN (Liu et al., 2019b) | 1k P | - | 92.9 | | | |
| DensePoint (Liu et al., 2019a) | 1k P | - | 93.2 | | | |
| PointASNL (Yan et al., 2020) | 1k P | - | 92.9 | | | |
| PosPool (Liu et al., 2020) | 5k P | - | 93.2 | | | |
| Point Trans. (Engel et al., 2020) | 1k P | - | 92.8 | | | |
| GBNet (Qiu et al., 2021b) | 1k P | 91.0 | 93.8 | 8.39M | 16.3 | 112 |
| GDANet (Xu et al., 2021b) | 1k P | - | 93.8 | 0.93M | 26.3 | 14.0 |
| PA-DGC (Xu et al., 2021a) | 1k P | - | 93.9 | | | |
| MLMSPT (Han et al., 2021) | 1k P | - | 92.9 | | | |
| PCT (Guo et al., 2021) | 1k P | - | 93.2 | | | |
| Point Trans. (Zhao et al., 2021) | 1k P | 90.6 | 93.7 | | | |
| CurveNet (Xiang et al., 2021) | 1k P | - | 94.2 | 2.04M | 20.8 | 15.0 |
| PointMLP **w/o vot.** | 1k P | 91.3 | 94.1 | 12.6M | 47.1 | 112 |
| PointMLP **w/ vot.** | 1k P | **91.4** | **94.5** | 12.6M | 47.1 | 112 |
| PointMLP-elite **w/o vot.** | 1k P | 90.9 | 93.6 | **0.68M** | 116 | 176 |
| PointMLP-elite **w/ vot.** | 1k P | 90.7 | 94.0 | **0.68M** | 116 | 176 |

we introduce a lightweight version of PointMLP named as *pointMLP-elite*, *with less than **0.7M** parameters and prominent inference speed (176 samples/second on ModelNet40 benchmark).*

Inspired by He et al. (2016); Hu et al. (2018), we present a bottleneck structure for the mapping function $\Phi_{pre}$ and $\Phi_{pos}$. We opt to reduce the channel number of the intermediate FC layer by a factor of $r$ and increase the channel number as the original feature map. This strategy is opposite to the design in Vaswani et al. (2017); Touvron et al. (2021) which increases the intermediate feature dimensions. Empirically, we do not observe a significant performance drop. This method reduce the parameters of residual MLP blocks from $2d^2$ to $\frac{2}{r}d^2$. By default, we set $r$ to 4 in PointMLP-elite. Besides, we also slightly adjust the network architecture, reducing both the MLP blocks and embedding dimension number (see appendix for details). Inspired by Xie et al. (2017), we also investigated a grouped FC operation in the network that divides one FC layer into $g$ groups of sub-FC layers, like group convolution layer. However, we empirically found that this strategy would largely hamper the performance. As a result, we did not consider it in our implementation.

## 4 EXPERIMENTS

In this section, we comprehensively evaluate PointMLP on several benchmarks. Detailed ablation studies demonstrate the effectiveness of PointMLP with both quantitative and qualitative analysis.

### 4.1 SHAPE CLASSIFICATION ON MODELNET40

We first evaluate PointMLP on the ModelNet40 (Wu et al., 2015) benchmark, which contains 9,843 training and 2,468 testing meshed CAD models belonging to 40 categories. Following the standard practice in the community, we report the class-average accuracy (mAcc) and overall accuracy (OA) on the testing set. We train all models for 300 epochs using SGD optimizer.

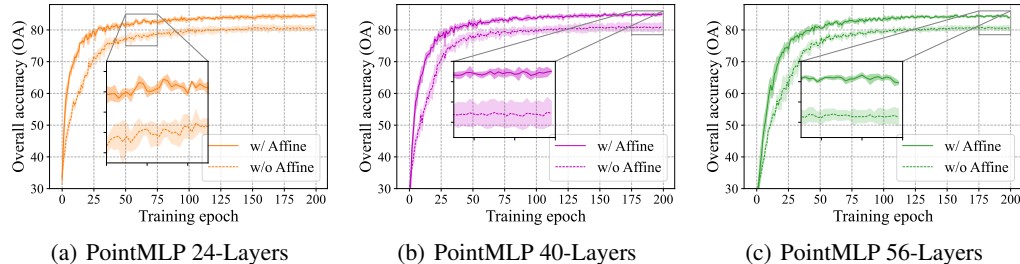

|  |  |  |
| :---: | :---: | :---: |
| (a) PointMLP 24-Layers | (b) PointMLP 40-Layers | (c) PointMLP 56-Layers |

Figure 3: Four run results (mean ± std) of PointMLP with/without our geometric affine module on ScanObjectNN test set. We zoom in on the details of PointMLP40 to show the stability difference.

Experimental results are presented in Table 2. Among these methods, our PointMLP clearly outperforms state-of-the-art method CurveNet by 0.3% (94.5% *vs.* 94.2%) overall accuracy with only 1k points. Note that this improvement could be considered as a promising achievement since the results on ModelNet40 recent methods have been saturated around 94% for a long time. Even without the voting strategy (Liu et al., 2019b), our PointMLP still performs on par or even better than other methods that are tested with voting strategy.

Despite having better accuracy, our method is much faster than the methods with sophisticated local geometric extractors. We compare PointMLP to several open-sourced methods and report the parameters, classification accuracy, training, and testing speed. As we stated previously, a key intuition behind this experiment is that model complexity can not directly reflect efficiency. For example, CurveNet is lightweight and delivers a strong result, whereas the inference cost is prohibitive **(15 samples/second)**. On the contrary, our PointMLP presents a high inference speed **(112 samples/second)**. To further reduce the model size and speed up the inference, we present a lightweight PointMLP-elite, which significantly reduces the number of parameters to 0.68M, while maintaining high-performance 90.9% mAcc and 94.0% OA on ModelNet40. With PointMLP-elite, we further speed up the inference to **176 samples/second**.

## 4.2 SHAPE CLASSIFICATION ON SCANOBJECTNN

While ModelNet40 is the de-facto canonical benchmark for point cloud analysis, it may not meet the requirement of modern methods due to its synthetic nature and the fast development of point cloud analysis. To this end, we also conduct experiments on the ScanObjectNN benchmark (Uy et al., 2019).

ScanObjectNN is a recently released point cloud benchmark that contains 15,000 objects that are categorized into 15 classes with 2,902 unique object instances in the real world. Due to the existence of background, noise, and occlusions, this benchmark poses significant challenges to existing point cloud analysis methods. We consider the hard-

Table 3: Classification results on ScanObjectNN dataset. We examine all methods on the most challenging variant (PB_T50_RS). For our pointMLP and PointMLP-elite, we train and test for four runs and report mean ± std results.

| Method | mAcc(%) | OA(%) |
| :--- | :---: | :---: |
| 3DmFV | 58.1 | 63 |
| PointNet (Qi et al., 2017a) | 63.4 | 68.2 |
| SpiderCNN (Xu et al., 2018) | 69.8 | 73.7 |
| PointNet++ (Qi et al., 2017b) | 75.4 | 77.9 |
| DGCNN (Wang et al., 2019) | 73.6 | 78.1 |
| PointCNN (Li et al., 2018b) | 75.1 | 78.5 |
| BGA-DGCNN (Uy et al., 2019) | 75.7 | 79.7 |
| BGA-PN++ (Uy et al., 2019) | 77.5 | 80.2 |
| DRNet (Qiu et al., 2021a) | 78.0 | 80.3 |
| GBNet (Qiu et al., 2021b) | 77.8 | 80.5 |
| SimpleView (Goyal et al., 2021) | - | 80.5±0.3 |
| PRANet (Cheng et al., 2021) | 79.1 | 82.1 |
| MVTN (Hamdi et al., 2021) | - | 82.8 |
| PointMLP (ours) | **83.9±0.5** | **85.4±0.3** |
| PointMLP-elite (ours) | **81.8±0.8** | **83.8±0.6** |

est perturbed variant (PB_T50_RS) in our experiments. We train our model using an SGD optimizer for 200 epochs with a batch size of 32. For a better illustration, we train and test our method for four runs and report the mean ± standard deviation in Table 3.

Table 4: Classification accuracy of pointMLP on ScanObjectNN test set using 24, 40, and 56 layers, respectively.

| Depth | mAcc(%) | OA(%) |
|---|---|---|
| 24 layers | 83.4±0.4 | 84.8±0.5 |
| 40 layers | **83.9±0.5** | **85.4±0.3** |
| 56 layers | 83.2±0.2 | 85.0±0.1 |

Table 5: Component ablation studies on ScanObjectNN test set.

| $\Phi_{pre}$ | $\Phi_{pos}$ | Affine | mAcc(%) | OA(%) |
|---|---|---|---|---|
| ✗ | ✓ | ✓ | 80.8±0.4 | 82.8±0.0 |
| ✓ | ✗ | ✓ | 83.3±0.3 | 84.7±0.2 |
| ✓ | ✓ | ✗ | 79.1±1.7 | 81.5±1.4 |
| ✓ | ✓ | ✓ | **83.9±0.5** | **85.4±0.3** |

Empirically, **our PointMLP surpasses all methods by a significant improvement on both class mean accuracy (mAcc) and the overall accuracy (OA)**. For example, we outperform PRANet by 4.8% mAcc and 3.3% OA. Even compared with the heavy multi-view projection method MVTN (12 views), our PointMLP still performs much better (85.39% 82.8%). *Notice that we achieve this by fewer training epochs and did not consider the voting strategy.* Moreover, we notice that our method achieves the smallest gap between class mean accuracy and overall accuracy. This phenomenon indicates that PointMLP did not bias to a particular category, showing decent robustness.

## 4.3 ABLATION STUDIES

**Network Depth.** Network depth has been exploited in many tasks but is rare in point cloud analysis. We first investigate the performance of PointMLP with different depths in Table 4. We vary the network depth by setting the number of homogeneous residual MLP blocks to 1, 2, and 3, respectively, resulting in 24, 40, and 56-layers PointMLP variants. Detailed depth formulation can be found in Appendix D. At first glance, we notice that simply increasing the depth would not always bring better performance; an appropriate depth would be a good solution. Additionally, the model gets stable with more layers introduced, as demonstrated by the decreasing standard deviation. When the depth is set to 40, we achieve the best tradeoff between accuracy and stability (85.4% mean accuracy and 0.3 standard deviations). Remarkably, PointMLP consistently achieves gratifying results that outperform recent methods, regardless of the depth.

**Geometric Affine Module.** Other work provides sophisticated local geometric extractors to explore geometric structures. Instead, our PointMLP discards these burdensome modules and introduces a lightweight geometric affine module. Figure 3 presents the results of PointMLP with/without the geometric affine module. By integrating the module, we systematically improve the performance of PointMLP by about 3% for all variants. The reasons for this large improvement are two-fold. First, the geometric affine module maps local input features to a normal distribution, which eases the training of PointMLP. Second, the geometric affine module implicitly encodes the local geometrical information by the channel-wise distance to local centroid and variance, remedying the deficiency of geometric information. Besides the gratifying improvements, the geometric affine module also largely boosts the stability of PointMLP, suggesting better robustness.

**Component ablation study.** Table 5 reports the results on ScanObjectNN of removing each individual component in PointMLP. Consistent with Figure 3, geometric affine module plays an important role in PointMLP, improving the base architecture by 3.9%. Remarkably, even without this module, which is an unfair setting for PointMLP, our base network stills achieves

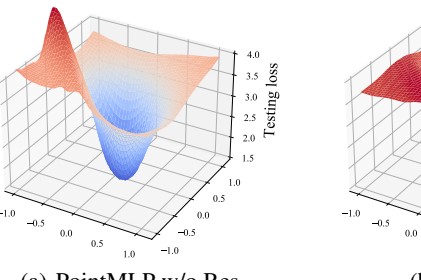 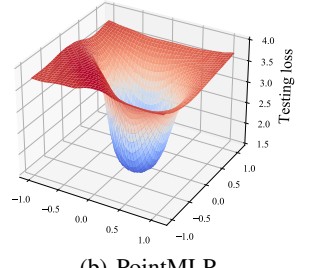

(a) PointMLP w/o Res.  (b) PointMLP

Figure 4: Loss landscape along two rand directions. By introducing residual connection, we ease the optimization of PointMLP and achieve a flat landscape like a simple shallow network intuitively.

Table 6: Part segmentation results on the ShapeNetPart dataset. Empirically, our method is much faster than the best method KPConv, and presents a competitive performance.

| Method | Cls. mIoU | Inst. mIoU | aero | bag | cap | car | chair | aerp-hone | guitar | knife | lamp | laptop | motor-bike | mug | pistol | rocket | skate-board | table |
|---|---|---|---|---|---|---|---|---|---|---|---|---|---|---|---|---|---|---|
| PointNet | 80.4 | 83.7 | 83.4 | 78.7 | 82.5 | 74.9 | 89.6 | 73.0 | 91.5 | 85.9 | 80.8 | 95.3 | 65.2 | 93.0 | 81.2 | 57.9 | 72.8 | 80.6 |
| PointNet++ | 81.9 | 85.1 | 82.4 | 79.0 | 87.7 | 77.3 | 90.8 | 71.8 | 91.0 | 85.9 | 83.7 | 95.3 | 71.6 | 94.1 | 81.3 | 58.7 | 76.4 | 82.6 |
| Kd-Net | - | 82.3 | 80.1 | 74.6 | 74.3 | 70.3 | 88.6 | 73.5 | 90.2 | 87.2 | 81.0 | 94.9 | 57.4 | 86.7 | 78.1 | 51.8 | 69.9 | 80.3 |
| SO-Net | - | 84.9 | 82.8 | 77.8 | 88.0 | 77.3 | 90.6 | 73.5 | 90.7 | 83.9 | 82.8 | 94.8 | 69.1 | 94.2 | 80.9 | 53.1 | 72.9 | 83.0 |
| PCNN | 81.8 | 85.1 | 82.4 | 80.1 | 85.5 | 79.5 | 90.8 | 73.2 | 91.3 | 86.0 | 85.0 | 95.7 | 73.2 | 94.8 | 83.3 | 51.0 | 75.0 | 81.8 |
| DGCNN | 82.3 | 85.2 | 84.0 | 83.4 | 86.7 | 77.8 | 90.6 | 74.7 | 91.2 | 87.5 | 82.8 | 95.7 | 66.3 | 94.9 | 81.1 | 63.5 | 74.5 | 82.6 |
| P2Sequence | - | 85.2 | 82.6 | 81.8 | 87.5 | 77.3 | 90.8 | 77.1 | 91.1 | 86.9 | 83.9 | 95.7 | 70.8 | 94.6 | 79.3 | 58.1 | 75.2 | 82.8 |
| PointCNN | 84.6 | 86.1 | 84.1 | 86.5 | 86.0 | 80.8 | 90.6 | 79.7 | 92.3 | 88.4 | 85.3 | 96.1 | 77.2 | 95.2 | 84.2 | 64.2 | 80.0 | 83.0 |
| PointASNL | - | 86.1 | 84.1 | 84.7 | 87.9 | 79.7 | 92.2 | 73.7 | 91.0 | 87.2 | 84.2 | 95.8 | 74.4 | 95.2 | 81.0 | 63.0 | 76.3 | 83.2 |
| RS-CNN | 84.0 | 86.2 | 83.5 | 84.8 | 88.8 | 79.6 | 91.2 | 81.1 | 91.6 | 88.4 | 86.0 | 96.0 | 73.7 | 94.1 | 83.4 | 60.5 | 77.7 | 83.6 |
| SynSpec | 82.0 | 84.7 | 81.6 | 81.7 | 81.9 | 75.2 | 90.2 | 74.9 | 93.0 | 86.1 | 84.7 | 95.6 | 66.7 | 92.7 | 81.6 | 60.6 | 82.9 | 82.1 |
| SPLATNet | 83.7 | 85.4 | 83.2 | 84.3 | 89.1 | 80.3 | 90.7 | 75.5 | 92.1 | 87.1 | 83.9 | 96.3 | 75.6 | 95.8 | 83.8 | 64.0 | 75.5 | 81.8 |
| SpiderCNN | 82.4 | 85.3 | 83.5 | 81.0 | 87.2 | 77.5 | 90.7 | 76.8 | 91.1 | 87.3 | 83.3 | 95.8 | 70.2 | 93.5 | 82.7 | 59.7 | 75.8 | 82.8 |
| KPConv | 85.1 | 86.4 | 84.6 | 86.3 | 87.2 | 81.1 | 91.1 | 77.8 | 92.6 | 88.4 | 82.7 | 96.2 | 78.1 | 95.8 | 85.4 | 69.0 | 82.0 | 83.6 |
| PA-DGC | 84.6 | 86.1 | 84.3 | 85.0 | 90.4 | 79.7 | 90.6 | 80.8 | 92.0 | 88.7 | 82.2 | 95.9 | 73.9 | 94.7 | 84.7 | 65.9 | 81.4 | 84.0 |
| PointMLP | 84.6 | 86.1 | 83.5 | 83.4 | 87.5 | 80.54 | 90.3 | 78.2 | 92.2 | 88.1 | 82.6 | 96.2 | 77.5 | 95.8 | 85.4 | 64.6 | 83.3 | 84.3 |

$81.5 \pm 1.4\%$ OA, outperforming most related methods (see Table 3). Removing $\Phi_{pre}$ function (MLPs before aggregator $\mathcal{A}$), the performance drops 2.6% overall accuracy. Combining all these components together, we achieve the best result 85.4% OA. See Appendix C for more ablations.

**Loss landscape.** We depict the 3D loss landscape (Li et al., 2018a) in Figure 4. Simply increasing the network depth may not achieve a better representation and even hamper the results. When removing the residual connection in PointMLP, the loss landscape turns sharp, and the performance plummets to 88.1% (6% drop) on ModelNet40. With residual connection, we greatly ease the optimization course of PointMLP and make it possible to train a deep network.

## 4.4 PART SEGMENTATION

Our PointMLP can also be gen-eralized to other 3D point cloud tasks. We next test PointMLP for 3D shape part segmentation task on the ShapeNetPart benchmark (Yi et al., 2016). The shapeNetPart dataset con-sists of 16,881 shapes with 16 classes belonging to 50 parts labels in total. In each class, the number of parts is between 2 and 6. We follow the set-tings from Qi et al. (2017b) that ran-domly select 2048 points as input for a fair comparison. We compare our methods with several recent works,

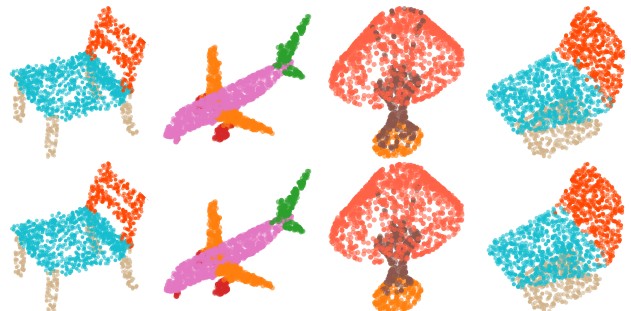

Figure 5: Part segmentation results on ShapeNetPart. Top line is ground truth and bottom line is our prediction.

including SyncSpecCNN (Yi et al., 2017), SPLATNet (Su et al., 2018), etc. We also visualize the segmentation ground truths and predictions in Figure 5. Intuitively, the predictions of our PointMLP are close to the ground truth. Best viewed in color.

## 5 CONCLUSION

In this paper, we propose a simple yet powerful architecture named PointMLP for point cloud analysis. The key insight behind PointMLP is that a sophisticated local geometric extractor may not be crucial for performance. We begin with representing local points with simple residual MLPs as they are permutation-invariant and straightforward. Then we introduce a lightweight geometric affine module to boost the performance. To improve efficiency further, we also introduce a lightweight counterpart, dubbed as PointMLP-elite. Experimental results have shown that PointMLP outper-forms related work on different benchmarks beyond simplicity and efficiency. We hope this novel idea will inspire the community to rethink the network design and local geometry in point cloud.

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

## A    POINTMLP DETAIL

We detail the architecture of PointMLP in Figure 6 (as well as PointMLP-elite in Figure 7) for a better understanding. Compared with PointMLP, the elite version mainly adjusts three configurations: 1) it reduces the number of residual point (Resp) MLP blocks; 2) it reduces the embedding dimension from 64 to 32, hence the overall model overhead is significantly alleviated; 3) by introducing a bottleneck structure, PointMLP further reduces the parameters by four times.

For part segmentation task, we use the framework presented in PointNet (Qi et al., 2017a) and replace the backbone to our PointMLP. With the only modification, we improve the performance from 85.1 to 86.1 Instance mIoU.

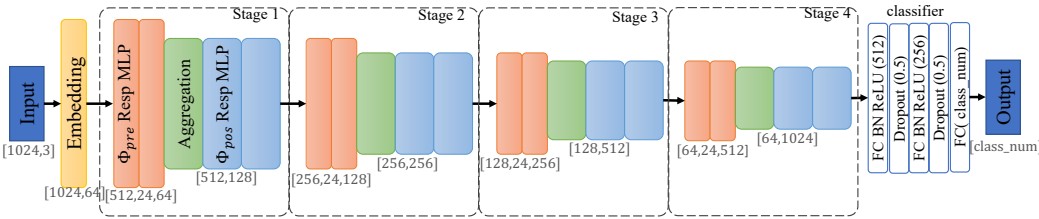

Figure 6: Detail architecture of PointMLP for classification.

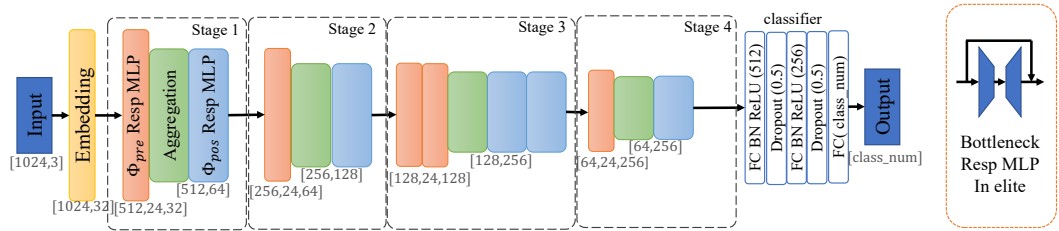

Figure 7: Detail architecture of PointMLP-elite for classification.

## B    DETAIL EXPERIMENTAL SETTING

### B.1    MODELNET40 AND SCANOBJECTNN

Our implementations are based on PyTorch. For ModelNet40, we train models for 300 epochs on one Tesla V100 GPU with a batch size of 32. All our models are trained using synchronous SGD with a Nesterov momentum of 0.9 and a weight decay of 0.0002. The learning rate is set to 0.1 initially. We use the cosine annealing scheduler (Loshchilov & Hutter, 2017) to adjust the learning rate. For each sample, we randomly select 1024 points and consider the same augmentation strategy as Qi et al. (2017b). The setting for ScanObjectNN is similar to ModelNet40, except we train all models for only 200 epochs.

For the reported speed in Table 2, we test the open-source code on a Tesla V100-pcie GPU. All the source codes we used are listed[2] in the footnote.

---

[2]all tested methods are listed bellow
PointNet++: https://github.com/erikwijmans/Pointnet2_PyTorch
CurveNet: https://github.com/tiangexiang/CurveNet
GBNet: https://github.com/ShiQiu0419/GBNet
GDANet: https://github.com/mutianxu/GDANet
PointConv: https://github.com/DylanWusee/pointconv
KPConv: https://github.com/HuguesTHOMAS/KPConv-PyTorch

## B.2 SHAPENETPART

Our setting for part segmentation task is following PointNet (Qi et al., 2017a). We randomly sample 2048 points for each sample and re-scale the input in a range of $[0.67, 1.5]$. Note that we did not test the result using a multi-scale testing strategy, which could further improve the performance, but is not realizable in real-world applications. Hence, we only report the single-scale results. Even the comparison is unfair, we still achieve competitive performance.

## C MORE DETAILED ABLATION STUDIES

*Skip connection.* Figure 4 shows the loss landscapes of our PointMLP with and without skip connections. We also consider adding skip connections to PointNet++ to validate the effectiveness of skip connections. Due to the structure of PointNet++, only two skip connections could be added without modifying the original architecture of PointNet++. By adding the skip connections, we achieve a classification accuracy of 92.7% on ModelNet40 in our re-implementation.

*Pre-MLP block vs. Pos-MLP block.* we also modified the configuration of our PointMLP and re-trained the model to investigate the importance of Pre-MLP and Pos-MLP blocks. In our original implementation, we set the pre-MLP block list to [2, 2, 2, 2] and the pos-MLP blocks list to $[2, 2, 2, 2]$. Here, we remove the pos-MLP blocks and change the pre-MLP blocks to $[4, 4, 4, 4]$ to match the block number. The 3-layer classifier can be considered as the MLP at the end of the last stage. We trained the models two times and got an average OA of 84.13% (83.87% and 84.39%), which is lower than vanilla PointMLP 85.4%, and even the result in Table 5 second-row 84.7%. This result indicates that pos-MLP does benefit our PointMLP, and simply adding more pre-MLP blocks does not help. We acknowledge that the effect of pos-MLP is not as strong as other components and believe that a detailed fine-tuning of the configurations would deliver an even better performance-efficiency balance.

*Geometric Affine Module Applications.* Geometric affine module plays an essential role in our PointMLP, exhibiting promising performance improvements. While this module can be considered as a plug-and-play method, the overlap with some local geometric extractors in other methods may limit its application. Here we integrate the module to two popular methods, PointNet++ and DGCNN, for illustration and experiment on the ModelNet40 benchmark. By integrating the geometric affine module, we improve the performance of PointNet++ to 93.3%, achieving an improvement of 1.4%. However, when integrating the module to DGCNN, we get a performance of 92.8%, which is slightly lower than the original results (92.9%). Note that both results are tested without voting.

## D POINTMLP DEPTH

Here we format the detailed formulation of layer number in our PointMLP. For the sake of clarity, we ignore Batch Normalization layers and activation functions. Let $\text{Pre}_i$ and $\text{Pos}_i$ indicate the repeating number of the $\Phi_{pre}$ block (which includes 3 layers) and $\Phi_{pos}$ block (which includes 2 layers) in $i$-th stage, respectively. Note that we have one layer in feature embedding in the beginning, one layer for channel number matching in each stage, and three layers in the classifier. Hence, the total number of learnable layers $L$ would be

$$L = 1 + \sum_{i=1}^{4} (1 + 2 \times \text{Pre}_i + 2 \times \text{Pos}_i) + 3.$$

As a result, the depth configuration of our network (24, 40, and 56) can be summarized as:

| Depth | $[\text{Pre}_1, \text{Pre}_2, \text{Pre}_3, \text{Pre}_4]$ | $[\text{Pos}_1, \text{Pos}_2, \text{Pos}_3, \text{Pos}_4]$ |
|---|---|---|
| 24 | $[1, 1, 1, 1]$ | $[1, 1, 1, 1]$ |
| 40 | $[2, 2, 2, 2]$ | $[2, 2, 2, 2]$ |
| 56 | $[3, 3, 3, 3]$ | $[3, 3, 3, 3]$ |

