# OpenReview forum: "Rethinking Network Design and Local Geometry in Point Cloud: A Simple Residual MLP Framework"
_ICLR.cc/2022/Conference — ICLR 2022 Poster_

### Official Review · Reviewer_8anY · 2021-11-01

**Correctness:** 3
**Technical Novelty And Significance:** 3
**Empirical Novelty And Significance:** Not applicable
**Recommendation:** 8
**Confidence:** 4

**Details Of Ethics Concerns:**

No issues.

**Main Review:**

The strength of this paper is the outperforming results and fast inference speed of the proposed method compared to most of the previous methods in various tasks, especially for a challenging task like classification with ScanObjectNN data. Also, the architecture seems very easy to implement from a PointNet++ code.

But, I also have several questions that I hope to be addressed by the authors in the rebuttal:

- What is the "voting" in the experiments? (E.g., w/o vot and w/ vot in Table 2)? I couldn't figure out what this voting meant.
- In Table 2, It seems the accuracy difference between CurveNet (Xiang et al., 2021) and the proposed method is marginal, while CurveNet is much lighter and faster. Also, the results of CurveNet seem to be without voting, so I guess there might be a chance for CurveNet to improve the accuracy if it uses the voting. What would be the other advantages of the proposed method compared to CurveNet? Would it be easier to implement? What happens if the proposed method is compared to CurveNet in the task of ScanObjectNN classification (Table 3) and ShapeNet classification (Table 6)?
- Why is the proposed method lighter and faster than PointNet++? Is it because of the residual connections? What happens if only the residual connections are used in PointNet++?
- It would also be great to see in Table 5 what happens when only one of the components in the columns on the left is used.
- Wouldn't it be possible to apply the geometric affine transformation (Equation 5) to any other architectures (compared in Table 1, for example)? It seems like this is the core module improving the performance.

The followings are also comments/questions about correction/clarification:
- What are the exact meanings of 24-Layers, 40-Layers, and 56-Layers in Figure 3 and Table 4? Is the number of MLP layers in "each" stage of the architecture shown in the appendix?
- Is having the post-MLP (\Phi_{pos}) the same as having a deeper pre-MLP in the intermediate stages, and adding an MLP at the end of the last stage? The effect of this post-MLP also seems marginal when seeing Table 5.
- In Equation 5, shouldn't the (f_{i,j} - f_i)^2 be a squared norm (\| f_{i,j} - f_i \|^2) so that the \sigma becomes a scalar? The text says that f_{i,j} and f_i are d-dimensional vectors.
- In Section 3.2. our PointMLP still performances efficiently -> our PointMLP still performs efficiently.
- In Equation 3, the symbol with a circle and a dot is not introduced in the text.


**Summary Of The Paper:**

This paper introduces a lightweight and fast neural network architecture processing 3D point cloud. While the idea is quite simple, the proposed architecture outperforms or matches the performance of the previous architectures in terms of both accuracy and also inference time with various tasks including ModelNet40 classification, ScanObjectNN classification, and ShapeNetPart segmentation, as shown in the experimental results. The main idea is to make two modifications in the PointNet++ architecture, 1) adding MLP layers after each of the point feature aggregation (max-pooling) steps, and 2) applying a geometric affine transformation to the features processed with an MLP in each small PointNet module. It's interesting to see that these small modifications (especially the second) make dramatic changes in the results.

**Summary Of The Review:**

I think this work introduces an interesting finding that small modifications in PointNet++ make dramatic changes in the results. But, more analyses would be needed to clearly see how each component in the proposed method (post-MLP, geometric affine, residual connections) affects the accuracy and inference time.

---

> ### Author Response · Authors · 2021-11-13
> **Response and proposed changes (1/3)**
>
> We thank R3 for helping improve our paper and appreciate that they recognized the novelty and value of our work. We address the concerns as follows and will revise the manuscript before the deadline.
>
> **Q1: What is voting?**
>
> The voting strategy is proposed in RS-CNN [1], which means we test with random scaling and average the predictions during the test. In recent years, voting strategy has been a widely considered strategy for point cloud classification methods in most papers. We explicitly followed the original implementation of the voting strategy and upload the voting codes to the anonymous link. We will add the citation in our revision.
>
> **Q2: In Table 2, It seems the accuracy difference between CurveNet ... compared to CurveNet in the task of ScanObjectNN classification and ShapeNet classification?**
>
> About the speed. As we stated in the caption of Table 2, we report the speed (not time) by samples/second, which means a higher value indicates faster speed. Compared with CurveNet, our PointMLP( and PointMLP-elite) trains **2.26-5.57**$\times$ faster, **7.46-11.7**$\times$ faster.
>
> About the performance. First, CurveNet and voting. CurveNet, which is a previous State-of-the-art result published in ICCV'21, is evaluated with the voting strategy to achieve 94.2\% classification accuracy. Without voting, the accuracy decreases to 93.8\% ( presented in the original paper). As a comparison, our PointMLP achieves 94.5\% and 94.1\% when evaluated with and without voting. Second, the accuracy differences on ModelNet40. We emphasize that 0.3\% improvements on ModelNet40 are promising improvements because the performance on ModelNet40 has been saturated around 94\% (93.8\%-94.2\%) for almost two years. As a reference, The link [2] collects state-of-the-art methods on ModelNet40, which shows the saturation intuitively. For the part-segmentation, CurveNet reported performance of 86.8\%, which is higher than our method. But we emphasized that this result is evaluated with the voting method, while we didn't consider it for the part-segmentation task. We also tested the CurveNet on the ScanObjectNN benchmark. Using their official model implementation, we got a result of 76.25\% mAcc and 80.08\% OA, which is much lower than our PointMLP (83.9\% mAcc and 85.4\%) and PointMLP-elite (81.8\%mAcc and 83.8\%). The codes, logs, and checkpoints are uploaded to the anonymous link.
>
>
> **Q3: Why is the proposed method lighter and faster than PointNet++?  What happens if only the residual connections are used in PointNet++?**
>
> This question may be related to the previous question about speed. Our method is slower than PointNet++ but much faster than all other competitors (see Table 1). Residual connections (i.e., skip connections) would not help the inference speed. We are slower than PointNet++ because our network is deeper and we introduce a geometric affine module. We are faster than others because the PointMLP is very simple and no sophisticated operations introduced. PointMLP-elite (0.68M) is lighter than PointNet++ (1.41M) because we reduce some layers and introduce a bottleneck structure. Section 3.4 and Appendix A demonstrate how we tailor the elite version of PointMLP.
> It is possible to add residual (skip) connections. Due to the structure of PointNet++, only two connections could be added, and the result on ModelNet40 is 92.7\% in our re-implementation. We upload the file to the anonymous link.
>
>
> **{Q4: Wouldn’t it be possible to apply the geometric affine ... the core module improving the performance.**
>
> It would be possible to apply the geometric affine module to other architectures. We integrate the module to two popular methods, PointNet++ and DGCNN, for illustration. Here is the result on ModelNet40: 93.3\% for PointNet++ with affine module and 92.8\% for DGCNN with the affine module. Note that both are tested without voting.

---

> > ### Author Response · Authors · 2021-11-13
> > **Response and proposed changes (2/3)**
> >
> > **Q5: What are the exact meanings of 24-Layers, 40-Layers, and 56-Layers in Figure 3 and Table 4?**
> > Here we format the detailed formulation of layer number in our PointMLP, ignoring Batch Normalization layers and activation functions.
> > Let $\mathrm{Pre}\_i$  and $\mathrm{Pos}\_i$  indicate the repeating number of the $\Phi_{pre}$ block (which includes 3 layers) and $\Phi_{pos}$ block (which includes 2 layers) in $i$-th stage, respectively. Note that we have 1 layer in feature embedding in the beginning, 1 layer for channel number matching in each stage, and 3 layer in the classifier. Hence, the total number would be
> > $1+ \sum_{i=1}^4\left(1+ 2\times \mathrm{Pre}\_i + 2\times \mathrm{Pos}\_i\right) + 3.$
> > Hence, the depth configuration of our network can be summarized as:
> >
> > | Layers | [$\mathrm{Pre}_1$, $\mathrm{Pre}_2$, $\mathrm{Pre}_3$, $\mathrm{Pre}_4$] | [$\mathrm{Pos}_1$, $\mathrm{Pos}_2$, $\mathrm{Pos}_3$, $\mathrm{Pos}_4$] |
> > |----|---|--|
> > | 24     | [1, 1, 1, 1]  | [1, 1, 1, 1]  |
> > | 40     | [2, 2, 2, 2] | [2, 2, 2, 2]  |
> > | 56     | [3, 3, 3, 3]   | [3, 3, 3, 3]  |
> >
> > Since our model (and the implementation) is flexible, we can easily change the configurations of the network as we want.
> >
> > **To be continued. We will respond to the rest concerns after we got the experimental results. Thanks for your kind patience.**
> >
> >
> >
> > **Reference**
> >
> > [1] Liu, Yongcheng, et al. "Relation-shape convolutional neural network for point cloud analysis."  CVPR, 2019.
> >
> > [2 ] 3d point cloud classification on modelnet40. https://paperswithcode.com/sota/3d-point-cloud-classification-on-modelnet40. Accessed: 2021-11-10.

---

> > > ### Author Response · Authors · 2021-11-19
> > > **Response and proposed changes (3/3)**
> > >
> > > **Q6:  Is having the post-MLP ($\phi_{pos}$) the same as having a deeper pre-MLP ... The effect of this post-MLP also seems marginal when seeing Table 5.**
> > >
> > > To answer this question, we modify the configuration of our PointMLP and retrained the model.
> > > In our original implementation, we set the pre-MLP block list to [2, 2, 2, 2] and the pos-MLP blocks list to [2, 2, 2, 2]. Here, we remove the pos-MLP blocks and change the pre-MLP blocks to [4, 4, 4, 4] to match the block number. The 3-layer classifier can be considered as the MLP at the end of the last stage. We trained the models two times and got an average OA of  **84.13**\%  (83.87\% and 84.39\%), which is lower than vanilla PointMLP 85.4\%, and even the result in Table 5 second row 84.7%. Codes/logs/models for this ablation study can be found in the anonymous link. This result indicates that pos-MLP does benefit our PointMLP and simply adding pre-MLP blocks does not help.
> > > We acknowledge that the effect of pos-MLP is not as strong as other components. We believe that a detailed fine-tuning of the configurations would deliver an even better performance-efficiency balance. We thank the reviewer for this constructive suggestion and will further explore the efficient PointMLP in our future work.
> > >
> > >
> > > **Q7: In Equation 5, shouldn't the $(f_{i,j} - f_i)ˆ{2}$ be a squared norm $|f_{i,j} - f_i|ˆ2$so that the $\sigma$ becomes a scalar? The text says that fi,j and fi are d-dimensional vectors.**
> > >
> > > We are sorry that this paragraph leads you misunderstood. $f_i\in R^d$ means the feature of $i$th selected point, and $f_{i,j}\in R^d$ indicates the $j$th neighbor of $f_i$. Hence, $(f_{i,j} - f_i)ˆ{2} \in R^d$ is d-dimensional vector (feature), as well as the $\sigma$.
> > > We are looking forward to a better presentation to reduce the confusion.
> > >
> > >
> > > **Q8 and Q9: typo and the lack of introduction to the symbol in Equation 3.**
> > >
> > > Thanks for pointing out this typo, we will fix it in our revision. "$\odot$" is a Hadamard product, as we stated in Figure 2. We will add an introduction to the text.

---

> > > > ### Comment · Reviewer_8anY · 2021-11-29
> > > > **Thank you for your response to my questions.**
> > > >
> > > > Thank you for your response to my questions. Most of my questions are well addressed by the comments. One of the major concerns that I and also R1 had was about the ablation study, especially for the geometric affine module --- what happens if this module is ablated, and what happens if the module is used in the other architectures. And, the authors addressed this concern well with additional experimental results in Sections 4.3 and B.3 in the supplementary.
> > > > While the technical novelty of this work may not be significant as mentioned by the other reviewers, and I think the paper is worth being presented in ICLR, so that many people can take the advantage of this simple yet powerful architecture in downstream applications. So, I keep my rating to 'accept'.

---

> > > > > ### Author Response · Authors · 2021-11-29
> > > > > **Thanks Reviewer 8anY for approving our work**
> > > > >
> > > > > Dear Reviewer 8anY,
> > > > >
> > > > > Thanks for agreeing that our response solves most of the concerns and our paper may help others in the future.
> > > > >
> > > > > In our future work, we would like to include more insightful investigations and deep analyses.
> > > > >
> > > > > Best,
> > > > > Authors

---

### Official Review · Reviewer_vGXQ · 2021-11-02

**Correctness:** 3
**Technical Novelty And Significance:** 2
**Empirical Novelty And Significance:** 2
**Recommendation:** 6
**Confidence:** 2

**Main Review:**

*Strengths*
- The paper is very well-written and easy to read.
- The architecture of PointMLP is simple for implementation. Since PointMLP is mainly based on pure residual MLPs, we can take advantage of the efficiency from the highly-optimized MLPs when training PointMLP.
- PointMLP outperforms previous SOTA models on ModelNet40 and ScanObjectNN.

*Weakness*
- As PointMLP is a refinement of PointNet++, I do not think the novelty of the idea is high.
- The performance of PointMLP is comparable with CurveNet on ModelNet40. In my opinion, the improvement by 0.3% is not significant.
- The results of the PointMLP in part segmentation task on ShapeNetPart are worse than the old architecture KPConv.

Some more comments and questions:
- Page 4, line 3: the local feature extraction of PointNet++ is formulated as
$$g_i = \mathcal{A}( \{ \Phi(f_{i,j})|j=1,...,K \} ).$$
- Similarly, in equation (4), the local feature extractor of PointMLP must be
$$g_i = \Phi_{pos}(\mathcal{A}( \{ \Phi_{pre}(f_{i,j})|j=1,...,K \} )).$$
- Why the small $\epsilon$ in equation (5) can stabilize the numerical computation?
- The authors claim that PointMLP is much faster than KPConv in part segmentation tasks. But I do not find anywhere in the paper nor the tables that support this claim.
- I am not sure if the performance of recent models based on sophisticated local geometric extractors is saturated or not as more and more efficient and accurate models of these kinds are still being investigated. Furthermore, as we see in the experiments, PointMLP does not outperform the previous SOTA model with significant margin (on ModelNet40 and ShapeNetPart). Therefore, stronger reasons are needed to support the claim that "detailed local geometrical information probably is not the key to point cloud analysis".

**Summary Of The Paper:**

The authors present a refinement of PointNet++, called PointMLP, in which the MLPs using in the local feature extractions in PointNet++ are replaced by residual MLPs. Together with a lightweight geometric affine module to stabilize the training, PointMLP outperforms previous models in the classification tasks on ModelNet40 and ScanObjectNN.

**Summary Of The Review:**

Since the idea of PointMLP is quiet simple and it can be considered as a refinement of PointNet++, I do not think that the novelty is very high, although the experiments show some interesting results.

---

> ### Author Response · Authors · 2021-11-13
> **Responses and proposed changes (1/2)**
>
> Thanks to R2 for helping improve our paper and thanks for all these constructive suggestions. We'll try to address the concerns, and revise our paper accordingly before the deadline,  in the hopes that you'll increase your evaluation of the paper. Please let us know if you think the response addressed your concerns and if any other points you want to discuss.
>
> **Q1: "As PointMLP is a refinement of PointNet++, I  do not think the novelty of the idea is high"**
>
>
> **First of all, we emphasize that PointMLP is not a simple improvement of PointNet++.** While PointMLP and PointNet++ are both based on MLP networks, they are essentially different. We summarized the similarities and differences in Table 1. Also, the architectures and designs are totally different. Moreover, in our PointMLP, we introduce skip connections, affine modules, configurable depth, and pos/pre-blocks. In the sense of design philosophy, we are more like to build a network like ResNet but based on the MLP framework.
>
> Second, as the title stated, the key point of this paper is to encourage the community to rethink the network design for point cloud analysis. While a number of sophisticated operations are proposed every year, we showcased that a simple MLP framework is able to achieve better performance (where our method rules the roost on both ModelNet40 and the challenging ScanObjectNN benchmark), while keeping a high inference speed.
>
> **We emphasize that our PointMLP is a novel method for point cloud analysis, In terms of both model architecture and design philosophy. Most important, we hope our PointMLP may help the community towards a better understanding of point cloud analysis.**
>
>
>
>
> **Q2: The performance of PointMLP ... 0.3\% on ModelNet40 is not significant.**
>
> We believe that 0.3\% improvements on the ModelNet40 benchmark are really promising.
> There are two reasons. First, the ModelNet40 is not a large-scale benchmark, even a little improvement is hard to achieve. That is the reason why we introduce the experiments on ScanObjectNN, a larger and more challenging benchmark. We outperform all previous work by a significant margin (over 3\%) on this benchmark.
>
> Second, the performance on ModelNet40 has been saturated around 94\% (93.8\%-94.2\%) for almost two years.
> For example, GBNet (first released in Nov 2019) [2] achieved a result of 93.8\%, and the current (Nov 2021) best result is CurveNet 94.2\%.
> As a reference, The link [1] collects state-of-the-art methods on ModelNet40, which shows the saturation intuitively. While 0.3\% may be numerically trivial, it indeed shows a noteworthy performance gap compared with all previous work.
>
>
> **Q3: The results ... worse than the old architecture KPConv.**
>
> We absolutely agree that our PointMLP performs slightly worse than KPConv. We can explain this from two aspects. First,  KPConv considers multi-scale grouping and employs the grid sampling strategies, which improve the performance a lot. However, the two methods greatly increase the model complexity and inference time. Second, KPConv uses the voting method for part segmentation tasks.
> on the contrary, we didn't employ any aforementioned methods because they are either time-consuming or infeasible in real-world applications. The reason is that we are targeting a simple, general, and feasible model for the point cloud analysis. Even under the unfair comparison, our method still achieves comparable performance and outperforms most methods.
> We thank the reviewer pointed out this issue, and we will emphasize it in the updated version.
>
>
> **Q4: Page 4, line 3: the local feature extraction ... Similarly, in equation (4) ...**
>
> Thanks a lot for the reviewer giving a better formulation. We will revise it.
>
> **Q5: Why the small value in equation (5) can stabilize the numerical computation?**
>
> This is a commonly used method in Mathematical Statistics. By adding the small value, we can avoid the situation where the divisor is 0, and hence helps the numerical stability. This method is also widely employed in normalization layers [3,4] and many machine learning methods [5].
>
>
>
> **Q6: The authors claim that PointMLP is much faster than KPConv ... support this claim.**
>
> The reason we didn't present the inference speed for part-segmentation is that each method has its own implementation details for the part-segmentation task besides the backbone, leading the comparison to be unfair. We claimed that our method is much faster than KPConv due to two factors: 1) as a backbone, we are faster than KPConv (see Table 2); 2) KPConv requires a pre-processing step, grid sampling, which increases the inference time a lot.
> From the KPConv paper, we found the testing speed is about 64 samples/second (where the pre-processing step is not considered).  In our test, PointMLP achieves a speed of 110.4 samples/seed.

---

> > ### Author Response · Authors · 2021-11-13
> > **Responses and proposed changes (2/2)**
> >
> > **Q7: I am not sure if the performance ... key to point cloud analysis**
> > First, the performances of PointMLP on ModelNet and ScanObjectNN are quite promising, even new methods are presented every year. The performance on ModelNet40 can be found in [1], which clearly shows the saturation issue as we discussed above. On the large and challenging real-world ScanObjectNN benchmark, we significantly outperform all previous work by a large margin (over 3\%), suggesting the superiority of our method.
> > Even under the unfair comparison, PointMLP still performs comparably with state-of-the-art methods on the ShapeNetPart benchmark.
> > As discussed in the Introduction section, more sophisticated extractors can not further improve the performance a lot and are hardly to be deployed in real-world applications. A novel perspective on point cloud analysis is required, like our PointMLP.
> >
> >
> > **Reference**
> >
> > [1] 3d point cloud classification on modelnet40. https://paperswithcode.com/sota/3d-point-cloud-classification-on-modelnet40. Accessed: 2021-11-10.
> >
> > [2] Qiu, Shi, Saeed Anwar, and Nick Barnes. "Geometric Feedback Network for Point Cloud Classification." CoRR (2019).
> >
> > [3] Ioffe, Sergey, and Christian Szegedy. "Batch normalization: Accelerating deep network training by reducing internal covariate shift." ICML, 2015.
> >
> > [4] Wu, Yuxin, and Kaiming He. "Group normalization."  ECCV, 2018.
> >
> > [5] Dixon, Wilfrid J., and Frank J. Massey Jr. "Introduction to statistical analysis." (1951).

---

> > > ### Comment · Reviewer_vGXQ · 2021-11-26
> > > **Thank you for your responses!**
> > >
> > > Thank you for your responses! My concern is the lack of novelty of the idea. Besides that, I still think that the results of PointMLP presented in this paper are not significantly strong to convince people to "rethink network design" as "detailed local geometrical information probably is not the key to point cloud analysis".
> > > However, based on other reviewers' comments and the authors' responses, I would like to increase my score to 6.

---

> > > > ### Author Response · Authors · 2021-11-29
> > > > **Thanks Reviewer vGXQ for approving our work**
> > > >
> > > > Dear Reviewer vGXQ,
> > > >
> > > > Thanks for your constructive suggestions and the score increase.
> > > >
> > > > We given this statement because our simple PointMLP achieves better performance than others (including concurrent works that released in recent days) without any sophisticated local geometric extactors. This is important in real-world point cloud applications.
> > > >
> > > > We hope our simple yet effective PointMLP would help the community towards a better understanding of point cloud analysis.
> > > >
> > > > Best,
> > > > Authors

---

### Official Review · Reviewer_7Y1Q · 2021-11-05

**Correctness:** 3
**Technical Novelty And Significance:** 2
**Empirical Novelty And Significance:** 2
**Recommendation:** 6
**Confidence:** 4

**Main Review:**

The proposed PointMLP achieves better accuracy while being simple and faster. This is not surprising. Empirically for point cloud processing I also found a lot of 'sophisticated' networks can be easily outbeat by much simple networks with tuned #layers and #channels.

My biggest concern for this paper is the lack of novelty and depth of this paper. It simply present the model and results, without telling why this simple model can outperform other delicately designed networks. This is more like a technical report.

Several claims made me confused:
1. 'even without the carefully designed operations for local geometry exploration': geometric affine module still uses local geometry. Did you compare PointMLP w/o geometric affine module with other models? I did not see any evidence that networks without local geometry exploration work better.
2. in part segmentation task, 'our method can achieve the best accuracy-speed tradeoff'. I did not see any speed comparison.

**Summary Of The Paper:**

This paper proposed an alternative point cloud feature extractor, named PointMLP. PointMLP is composed of residual MLPs and geometric affine modules. Classification results on ModelNet40 dataset show the proposed methods achieves slightly better accuracy while using much smaller number of parameters and faster runtime. The proposed method achieves better accuracy in classification results on ScanObjectNN dataset and is on par with state-of-the-art methods for 3D shape part segmentation task on the ShapeNetPart benchmark.

**Summary Of The Review:**

My biggest concern for this paper is the lack of novelty and depth of this paper. I still give a positive review because I got sick of papers that boast about their fancy design but in reality cannot outperform a simple network.


==========================================

Based on other reviewers' comments and the authors' responses, I would like to keep my original score.

---

> ### Author Response · Authors · 2021-11-13
> **Responses and proposed changes**
>
> Thanks to R1 for helping improve our paper! We are delighted that reviewer pointed out this dilemma of point cloud analysis, which motivates us to conduct this work. Hope the following responses can answer your questions and increase your evaluation.
>
>
>
> **Q1: geometric affine module still uses local geometry ...   I did not see any evidence that networks...**
>
> We do agree with the reviewer that our geometric affine module utilized the local geometric information. However, we emphasize that 1) acknowledge the contributions of other local geometric extractors, but a more succinct framework should be studied considering both the efficiency and accuracy; 2) our affine operation is extremely cheap (only mean value and the standard deviation are required) and the performance is promising.
>
> Figure 3 and Table 5 present the results of removing the affine module on ScanObjectNN. Removing the geometric affine module, the performance decreased to 81.5$\pm$1.4\% OA  (original result is 85.4$\pm$0.3\%). This result is still better than most methods, as shown in Table3.  For your convenience, we list the results on ScanObjectNN in the following table.
>
> | Method | mAcc(%) | OA(%) |
> |---|---|----|
> | 3DmFV | 58.1 | 63 |
> | PointNet (2017 CVPR) | 63.4 | 68.2 |
> | SpiderCNN (2018 ECCV) | 69.8 | 73.7 |
> | PointNet++ (2017 NeurIPS) | 75.4 | 77.9 |
> | DGCNN (2019 TOG) | 73.6 | 78.1 |
> | BGA-DGCNN (2019 ICCV) | 75.7 | 79.7 |
> | BGA-PN++ (2019 ICCV) | 77.5 | 80.2 |
> | DRNet (2021 WACV) | 78.0 | 80.3 |
> | GBNet (2021 TMM) | 77.8 | 80.5 |
> | Simple View (2021 ICML) | - | 80.5$\pm$0.3 |
> | **PointMLP w/o Affine** | **79.1$\pm$1.7** | **81.5$\pm$1.4** |
> | PRANet (2021 TIP) | 79.1 | 82.1 |
> | MVTN-12view (2021 ICCV) | - | 82.8 |
> | **PointMLP** | **83.9$\pm$0.5** | **85.4$\pm$0.3** |
> | **PointMLP-elite** | **81.8$\pm$0.8** | **83.8$\pm$0.6** |
>
>
> Even without the geometric affine module, our incomplete  PointMLP still performs better than most methods. Introducing affine module would largely improve the performance and stabilize the model.
>
> **Q2: In part segmentation task ... did not see any speed comparison.**
>
> The reason we didn't present the inference speed for part-segmentation is that each method has its own implementation details for the part-segmentation task besides the backbone, leading the comparison to be unfair. We claimed that our method achieve the best accuracy-speed tradeoff due to the factor that our method performs much faster than others as a backbone. Thanks for reviewer points out this, and we will revise the statement to avoid overclaiming.

---

> ### Author Response · Authors · 2021-11-29
> **Thanks Reviewer 7Y1Q for approving our work**
>
> Dear Reviewer 7Y1Q,
>
> Thanks for agreeing with the idea and motivation of our work.
>
> Besides, we appreciate the suggestions for additional experiments (added in the appendix) to improve our work.
>
> Best,
> Authors

---

### Author Response · Authors · 2021-11-24
**Summary of the Revision**

We thank all reviewers for their insightful reviews and suggestions. Here, we provide a summary of the updates (marked in red color) made in the new version. In addition, we respond to each reviewer separately regarding points particular to their review.

- The following updates have been incorporated in the main manuscript:

- We revised Equation (1) and Equation (4) as suggested by Reviewer 2.

- We added an introduction to the $\odot$ as suggested by Reviewer 3.

- In Section 3.3, we slightly updated the presentation to better introduce the geometric affine module, avoiding confusion.

- In Section 4.1, we explained and emphasized the improvement on ModelNet40, discussing the results with and without the voting strategy.

- In Section 4.4, we updated the presentation of accuracy-speed trade-off to avoid overclaim as suggested by Reviewer 1.

- We added more ablation studies in the appendix, including skip connection, Pre-MLP block vs. Pos-MLP block, and geometric affine module applications.

- We updated a typo as pointed by Reviewer 3.

- We also presented the detailed calculations of PointMLP depth in the appendix.

We hope that our response and revision address your concerns and questions. We are happy to have further discussions if you have any additional concerns or comments.

---

### Public Comment · ~Weiwei_Sun3 · 2022-04-04
**What's role of local normalization layer (GEOMETRIC AFFINE MODULE)**

Dear authors,

Thanks for this interesting work!!

This paper reveals an interesting finding -- existing works were fancy, and complex but sadly didn't work well. And this paper presents a simple yet effective framework. I really appreciate this kinda work.

However, maybe I missed something. It seems to me that the paper didn't analyze why the proposed simple baseline outperforms all the previous works. If I understand the core component correctly, I feel the main difference is that this paper utilizes the local normalization layer (GEOMETRIC AFFINE MODULE). This is a bit surprising.  Since we actually also studied the impact of different normalization layers (e.g., CNe, ACNe), we didn't observe this large improvement. Maybe we used the normalization in different ways. So I wonder what's the key to the success of the proposed geometric affine module compared with the normalization layers in [1, 2].

I would appreciate it very much if the author can reply.

1, CNe:Learning to Find Good Correspondences
2, ACNe: ACNe: Attentive Context Normalization for Robust Permutation-Equivariant Learning

---

> ### Public Comment · ~Xu_Ma2 · 2022-04-04
> **Response**
>
> Hi,
>
> Thanks a lot for your interest!
>
> First (for the simple baseline), we notice that even simple designs can achieve competitive results compared with some sophisticated designs. This phenomenon is demonstrated in [1] and Figure.3/ Table.3 (removing this module, the baseline still achieves promising performance), indicating that sophisticated designs may be unnecessary.
>
> Second (for the module), yes, this module is very close to the normalization layer. The motivation behind this design is that we are trying to replace the local aggregators with simple yet effective operations, and a normalization-like operation perfectly fits these requirements. Note that the Geometric Affine Module is not a simple BN or LN; see: https://github.com/ma-xu/pointMLP-pytorch/blob/c1d6235405a8e53027d5afa1349a368788fa2469/classification_ScanObjectNN/models/pointmlp.py#L177
>
> Last (for the references), thanks for the two references. We haven't read these two papers and cannot tell the differences (I assume they are different) and the reason for the performance gap. We will read these papers recently to look into details.
>
> Please let me know if you have any further questions or concerns.
>
> Best,
> Xu
>
>
> [1] Liu, Ze, et al. "A closer look at local aggregation operators in point cloud analysis." European Conference on Computer Vision. Springer, Cham, 2020.

---

> > ### Public Comment · ~Weiwei_Sun3 · 2022-04-04
> > **Thanks for the quick reply**
> >
> > Thanks a lot for the quick reply. Looking forward to your official presentation and your more insights regarding the idea behind this simple design.
> > All the best!

---

### Decision · Program_Chairs · 2022-01-20

**Decision:**

Accept (Poster)

**Comment:**

This paper proposes a new architecture for point cloud processing, with good empirical results. All reviewers recommended accept. AC does not see a reason to overturn the consensus.